# Natural Products as Modulators of Iron Metabolism and Ferroptosis in Diabetes and Its Complications

**DOI:** 10.3390/nu17162714

**Published:** 2025-08-21

**Authors:** Yuanfen Xie, Chunqin Li, Xige Dong, Beilei Wang, Jiaxin Qin, Huanhuan Lv

**Affiliations:** 1School of Life Sciences, Northwestern Polytechnical University, Xi’an 710072, China; 2579770383@mail.nwpu.edu.cn (Y.X.); lichunqin@mail.nwpu.edu.cn (C.L.); dxg@mail.nwpu.edu.cn (X.D.); wangbl@mail.nwpu.edu.cn (B.W.); qinjiaxin@mail.nwpu.edu.cn (J.Q.); 2Key Laboratory for Space Bioscience and Biotechnology, Northwestern Polytechnical University, Xi’an 710072, China

**Keywords:** diabetes, iron metabolism, ferroptosis, diabetic complications, natural products, traditional Chinese medicine (TCM)

## Abstract

Diabetes, a major global healthcare challenge, is characterized by chronic hyperglycemia and significantly exacerbates the severity of systemic complications. Iron, an essential element ubiquitously present in biological systems, is involved in many biological processes facilitating cell proliferation and growth. However, excessive iron accumulation promotes oxidative damage through the Fenton reaction, thereby increasing the incidence of diabetes and worsening diabetic complications. Notably, ferroptosis, an iron-dependent form of regulated cell death driven by lipid peroxidation, has emerged as a key mechanism underlying diabetes and diabetic complications. In this review, we provide an update on the current understanding of iron metabolism dysregulation in diabetes risk, and disclose the mechanistic links between iron overload and diabetes evidenced in hereditary hemochromatosis and thalassemia. We particularly highlight iron-mediated oxidative stress as a central nexus impairing glucose metabolism and insulin sensitivity. Furthermore, we discuss the significance of dysmetabolic iron and ferroptosis activation in the progression of diabetes and diabetic complications, as well as the possible application of natural products for iron metabolism regulation and ferroptosis-inhibition-targeted therapeutic strategies to treat diabetes and diabetic complications.

## 1. Introduction

Diabetes is one of the fastest-growing diseases globally, with an estimated 693 million adults affected by 2045 [1]. This metabolic disorder manifests through chronic hyperglycemia, multifaced complications, and impaired glucose and lipid metabolism. The social and economic burden of diabetes is staggering, accounting for approximately 12% of the global health expenditure and leading to significant losses in productivity due to disability and premature mortality [2]. Understanding the underlying mechanisms and developing effective treatments are thus of utmost importance.

Iron homeostasis has been implicated as a pathogenic mediator in diabetes pathogenesis. Epidemiological evidence links iron to diabetes, with higher iron levels increasing type 2 diabetes risk and depletion reducing it [3]. Clinical observation shows elevated ferritin levels in diabetic patients [4]. The direct effects of iron overload on diabetes are evidenced on hereditary hemochromatosis and thalassemia [5]. Iron exerts cytotoxicity on pancreatic β cells through Fenton reaction-derived oxidative stress and mitochondrial dysfunction, contributing to diabetes risk. It also regulates metabolism and fuel choice [6].

The relationship between iron and diabetes is complex and multifactorial. Beyond its direct effects on pancreatic β cells and insulin resistance, iron also influences other cellular processes that contribute to diabetes development and progression. Ferroptosis, an iron-dependent form of cell death driven by lipid peroxidation accumulation, has recently emerged as a novel pathogenic mechanism in diabetes [7,8]. Evidence suggests that ferroptosis may play a role in diabetes pathogenesis and its complications, including nephropathy, retinopathy, and cardiovascular diseases [9,10]. While iron itself can directly impair β cell function and insulin resistance through mechanisms such as oxidative stress, inflammation, and mitochondrial dysfunction, the dysregulated iron metabolism and lipid peroxidation associated with ferroptosis further exacerbate these effects by promoting cellular damage and organ dysfunction [8]. Understanding the interplay between ferroptosis and diabetes could pave the way for novel therapeutic strategies to mitigate disease progression and its associated complications.

Although the iron-related mechanisms in diabetes are increasingly understood, current treatments still fall short in effectively managing the disease and its complications. Thus, traditional Chinese medicine (TCM) and TCM-derived natural products have gained significant attention as alternative or complementary approaches for managing diabetes and its complications [11]. Recent studies indicate that certain TCM formulations and bioactive compounds derived from natural sources may exert therapeutic effects by regulating iron metabolism and ferroptosis signaling [12,13]. These findings highlight the potential of TCM and natural products in addressing the multifaceted nature of diabetes, particularly through their ability to modulate cellular processes such as iron homeostasis and lipid peroxidation.

In this review, we first highlight recent research progress on the epidemiological and molecular evidence linking iron dysregulation to diabetes pathogenesis. Subsequently, we analyze how iron-mediated oxidative stress impairs glucose metabolism and insulin sensitivity. A dedicated focus explores ferroptosis activation in the development of diabetes and diabetic complications. Finally, we summarize the evidence supporting the potential of TCM and natural products targeting iron metabolism and ferroptosis as improved treatments for diabetes and its complications.

A comprehensive systematic literature search was conducted across major international databases (Web of Science and PubMed/MEDLINE) using an expanded set of search terms, including “diabetes mellitus,” “T2DM,” “type 2 diabetes,” “type 1 diabetes,” “DM,” “diabetic complications,” “diabetic neuropathy,” “diabetic retinopathy,” “diabetic nephropathy,” “iron overload,” “iron dysregulation,” “ferroptosis,” “hepcidin,” “ferritin,” “transferrin,” “iron homeostasis,” “iron chelation,” “labile iron pool,” “cellular iron,” “phytochemicals,” “herbal medicine,” “polyphenols,” “botanical extracts,” “plant-derived compounds,” “natural bioactive compounds,” and “nutraceuticals”. Boolean operators (AND/OR) were employed to link these concept clusters, with additional manual searching of reference lists and recent reviews to identify potentially relevant studies that might have been missed in the initial database search.

## 2. Iron–Diabetes Pathophysiological Interplay

To establish a clear understanding of iron’s role in diabetes, the following section is structured from the basic to the special, from phenomena to mechanisms. First of all, the general phenomenon of iron metabolism dysregulation in diabetes is expounded. Then, the correlations between iron and different types of diabetes are respectively discussed. Subsequently, the relationships between diabetes and diseases related to iron overload are focused on. Finally, the mechanisms of iron in diabetes are thoroughly studied, including the impact on glycemic control and the roles of iron status and iron metabolism related genes in insulin regulation.

### 2.1. Dysregulated Iron Homeostasis in Diabetes

Epidemiological evidence demonstrates a strong association between iron and diabetes. Higher systemic iron level is linked to an elevated risk of type 2 diabetes, whereas iron depletion is associated with a reduced risk [3]. In diabetic patients, systemic iron status is altered, with increased levels of serum iron, transferrin saturation, and ferritin, as well as decreased transferrin level. Serum hepcidin level is higher in diabetic patients compared to age-matched controls [4].

In studies examining iron-related parameters in diabetic patients, specific findings have emerged. Iron-binding antioxidant capacity (IB-AC) is significantly lower in the plasma of diabetic subjects [14]. It should be noted that the concentrations of acute-phase proteins—including ceruloplasmin (CP), transferrin, and albumin—may not reliably indicate iron metabolism in the context of diabetes-associated chronic inflammation and oxidative stress. Additionally, non-transferrin-bound iron (NTBI) is commonly present in type 2 diabetic patients, with a higher prevalence in advanced diabetes patients. NTBI level is strongly correlated with diabetes severity, highlighting its potential role in diabetes-related complications [15].

The disruption of iron metabolism in diabetes also extends to gene expression changes. In patients with type 2 diabetes, the expression of ferroportin (FPN) and solute carrier family 11 member (SLC11A2) is increased in the intestine, associated with elevated iron stores and serum hepcidin levels [16]. In streptozotocin (STZ)-induced diabetic mice, the mRNA expression of the iron importer divalent metal transporter 1 (DMT1) is significantly increased in the duodenum [17]. Moreover, six-transmembrane epithelial antigen of the prostate 3 (STEAP3) is upregulated in diabetic wounds [18], and feline leukemia virus subgroup C receptor 1 (FLVCR1) mRNA and protein levels are increased in adipose tissue in patients with type 2 diabetes [19]. Zrt/Irt-like protein 14 (ZIP14), a transport protein, is highly expressed in rats with diabetic nephropathy and in human kidney proximal tubular cell lines under high glucose conditions [20].

These alterations in gene expression are correlated with the risk of diabetes. For example, transferrin receptor (TFRC), hemochromatosis (HFE), and heme oxygenase 1 (HMOX1) are associated with an increased risk of diabetes, while transmembrane protease, serine (TMPRS), and mothers against decapentaplegic homolog 7 (SMAD7) are associated with a decreased risk [21]. Overall, the complex interplay between iron metabolism disruption, gene expression changes, and the risk of diabetes highlights the importance of understanding these mechanisms for potential therapeutic interventions.

### 2.2. Iron’s Association with Different Types of Diabetes

#### 2.2.1. Iron and Type 1 Diabetes

Type 1 diabetes is one of the most common endocrine and metabolic conditions in children [22]. In the Norwegian Mother and Child Cohort (MoBa), a large-scale study involving 94,209 pregnancies revealed that 373 children develop type 1 diabetes by the end of follow-up. The incidence rate is significantly higher among children exposed to maternal iron supplementation (36.8 cases per 100,000 children per year) compared to those unexposed (28.6 cases per 100,000) [23]. These findings suggest that type 1 diabetes may be influenced by prenatal exposures such as maternal iron supplementation. Another study also shows that the mean iron concentration in the serum of newly diagnosed type 1 diabetes patients is lower than that in those who have had diabetes for a longer time [24]. Therefore, there is often iron-deficiency anemia occurring in type 1 diabetes. Type 1 diabetes is associated with chronic low-grade inflammation, which may increase hepcidin synthesis. However, a cross-sectional study reveals that there is no difference on hepcidin concentration between functional iron deficiency in children with type 1 diabetes and normal iron status [25].

#### 2.2.2. Iron and Type 2 Diabetes

Type 2 diabetes is a progressive metabolic disorder characterized by insulin resistance and pancreatic β cell dysfunction. Ferritin and transferrin saturation are associated with increased risk of type 2 diabetes [26]. Patients with newly diagnosed type 2 diabetes exhibit significantly higher serum ferritin levels compared to healthy controls. Additionally, male patients show elevated serum iron and transferrin saturation levels relative to female patients [27]. Patients with type 2 diabetes have reduced serum hepcidin level and increased circulating iron and ferritin levels [28]. Higher heme iron intake is associated with a higher risk of type 2 diabetes [29]. Therefore, type 2 diabetes is closely associated with iron metabolism. Patients with this condition frequently exhibit iron overload, as evidenced by elevated levels of serum ferritin, serum iron, and transferrin saturation. Nevertheless, in specific circumstances, iron deficiency anemia may also develop [30].

#### 2.2.3. Iron and Gestational Diabetes

Gestational diabetes is the most frequently occurring complication during pregnancy. It is a cause of concern due to increased risks on both mother and fetus. Serum ferritin concentration and oxidative status index in a gestational diabetes group were higher than controls [31]. A moderate fructose iron-enriched intake increases the risk of glucose disorder during pregnancy and oxidative damage on newborns [32]. Iron supplement-induced oxidative stress further influences insulin resistance in pregnant women with gestational diabetes [33]. Dysregulated oxidative stress contributes to reduced placental iron transport under gestational diabetes-mediated hyperglycemic conditions [34]. High glucose increases reactive oxygen species (ROS), malondialdehyde (MDA), and iron content, while decreasing glutathione (GSH) and glutathione peroxidase 4 (GPX4) in HTR-8/SV neo cells; this indicates the involvement of ferroptosis in the gestational diabetes [35]. However, a meta-analysis shows that there is no difference in total iron binding capacity (TIBC) and transferrin concentration in women with and without gestational diabetes, as well as no association among the increased transferrin receptor, increased intake of dietary iron, and the increased odds ratio for gestational diabetes [36].

### 2.3. Diabetes in Hereditary Iron Overload Disorders—Hereditary Hemochromatosis and Thalassemia

#### 2.3.1. Hereditary Hemochromatosis-Associated Diabetes

Hereditary hemochromatosis is a heterogeneous genetic disorder that results in iron overload in tissue from unregulated and excessive intestinal iron absorption and the release of iron from senescent erythrocytes [37]. Hereditary hemochromatosis is caused by mutations in iron regulatory genes, including *HAMP* (hepcidin gene), *HFE* (hemochromatosis gene), *TFRC* (transferrin receptor), *HJV* (hemojuvelin), and *SLC40A1* (ferroportin), and heterozygous mutations in BMP6 pro-peptide [38,39,40,41]. Adult humans with iron overload from hereditary hemochromatosis have high prevalence of diabetes and impaired glucose tolerance. The impaired glucose tolerance in hereditary hemochromatosis is associated with decreased insulin secretion capacity. Paradoxically, there is no consistent association between diabetes and those with *HFE* mutations [42]. Haemochromatosis patients with diabetes exhibited further declines in acute insulin response to glucose and insulin resistance [43].

#### 2.3.2. Thalassemia-Associated Diabetes

Thalassemia is a heterogeneous genetic disorder that resulted in ineffective erythropoiesis, increased hemolysis, and dysregulated iron homeostasis through the unbalanced globin chain production [44]. Clinically, thalassemia manifests as two forms: α-thalassemia and β-thalassemia. Diabetes is a common complication found in β-thalassemia patients. A cross-sectional study showed patients with β-thalassemia having increased fast plasma glucose, fast insulin and insulin resistance index, and decreased insulin sensitivity index [45].

### 2.4. Iron-Mediated Oxidative Stress in Diabetes Pathogenesis

Within the body iron pool, it is generally considered that iron that is not sequestered on transferrin or bound to other iron-binding proteins, referred to as NTBI, contains a proportion of redox active iron capable of inducing oxidative damage to cells and tissues. Increased lipid peroxidation contributes to diabetic complications, and redox-active iron is known to play an important role in catalyzing peroxidation reactions [46]. Iron-mediated oxidative stress may be a mechanism linking poor glycemic control with vascular dysfunction in type 2 diabetes [47]. Increased iron store is significantly associated with the development of type 2 diabetes by damaging the pancreatic β cells and causing insulin resistance.

### 2.5. The Impact of Iron Homeostasis on Glycemic Control and Insulin Regulation

#### 2.5.1. Impacts of Iron Level Variations on Glycemic Control

Increased hepatic glucose production is a risk for diabetes with excess iron in tissues. Dysregulation of hepatic glucose production and changes in the capacity to metabolize different fuels may predispose to diabetes in the condition of iron overload. A cross-sectional study suggests that high serum ferritin concentrations are linked to impaired glucose homeostasis in patients with metabolic syndrome [48].

Hepatic gluconeogenesis acts to maintain glucose supply during starvation. In diabetes, it is inappropriately active and contributes to hyperglycemia. Iron treatment increases cellular glucose intake and inhibits gluconeogenesis in rat liver-derived cells [49]. Mice fed with high-iron diets exhibit elevated adenosine monophosphate-activated protein kinase (AMPK) activity, enhanced glucose uptake, suppressed glucose output, and impaired insulin signaling in skeletal muscle and liver [50]. High-iron diet induces less hepatic glycogen deposition; downregulated glucose transporter type 4 (GLUT4) expression in the liver; and increased fasting blood glucose, serum insulin, and glycated hemoglobin A1c (GHbA1c) in *db/db* mice [51]. Hepatocytes from *Hamp*^−/−^ mice show greater sensitivity to the effects of insulin, evidenced by higher phosphorylation level of protein kinase B (Akt) and glycogen synthase kinase 3β (GSK 3β) [52]. *Hfe*^−/−^ mice exhibit increased glucose uptake in skeletal muscle, decreased glucose oxidation and increased fatty acid oxidation in skeletal and cardiac muscle, and increased hepatic glucose output [6].

Iron deficiency also impairs glucose homeostasis and may negatively affect glycemic control [53]. Iron-deficient rats show increased peripheral glucose uptake in response to insulin. Hepcidin is a gluconeogenic sensor in mice during starvation. Activation of hepcidin and perturbation of iron homeostasis during starvation-induced gluconeogenesis seem to represent a general defensive response in rodents [54]. The activation of hepcidin disturbs mitochondrial function and increases gluconeogenesis [55].

#### 2.5.2. Iron Regulation in Pancreatic β Cell Function and Insulin Sensitivity

Pancreatic β cell failure is a precondition in the pathogenesis of diabetes. Research has shown that pancreatic β cell function is negatively associated with serum iron, ferritin, and transferrin saturation and positively associated with transferrin. Insulin sensitivity is positively related to serum ferritin and transferrin saturation and negatively related to transferrin [27]. However, a different result shows that serum ferritin is negatively associated with pancreatic β cell function that is independent of insulin sensitivity [56].

Iron metabolism in pancreatic β cells is complex. Pancreatic β cells express hepcidin, which links iron metabolism to glucose sensing [57]. Given the susceptibility of pancreas to iron-related disorders such as hemochromatosis and secondary iron overload, alterations in iron metabolism have significant impacts on the pancreas [58]. Iron deficiency by *Irp2* knockout in pancreatic β cells impairs glucose intolerance via insulin secretion defects, which may have relevance to type 2 diabetes [59]. *Heph* (Hephaestin) and *Cp*, multicopper ferroxidases that oxidize Fe^2+^ to Fe^3+^, are known to facilitate iron efflux with iron exporter FPN1. Single *Heph* or *Cp* knockout has a limited effect on pancreatic iron level, while double knockout of *Heph* and *Cp* exaggerates pancreatic iron deposition with oxidative damage [60,61]. *Bmp*^−/−^ mice exhibit the feature of age-dependent iron accumulation in the exocrine pancreas [62].

Excessive iron deposition results in impaired function of the endocrine and exocrine pancreas. Iron overload with ferric ammonium citrate (FAC) or iron dextrose induces MIN6 cell dysfunction and ferroptosis of islet β cells [63]. Iron accumulation causes pancreatic β cell oxidative damage, and further leads to ferroptosis under diabetic conditions [64]. Pancreatic β cells with low expression of antioxidant enzymes are susceptible to ferroptosis. High iron exposure resulted in decreased insulin secretion and cellular insulin content in MIN6 cells [65]. Excess iron in pancreatic β cells impairs mitochondrial function and glucose-stimulated insulin secretion [43,66,67]. Reduction in iron level by deferiprone (DFP) decreases mitochondrial iron content and ROS level and restores glucose-stimulate insulin secretion in NAF-1-repressed INS-1E pancreatic cells [68].

Pro-inflammatory cytokines are cytotoxic to pancreatic β cells, as they impair their function and activate the intrinsic apoptotic pathway [69]. They also have an impact on iron metabolism in pancreatic β cells. The proinflammatory cytokine interleukin-1β (IL-1β) induces DMT1 expression correlating with increased iron content and ROS production in pancreatic β cells [70]. Alternatively, alteration of iron level affects the effects of cytokines on pancreatic β cells. DMT1 silence protects pancreatic β cells against IL-1β-mediated cell death [71]. FAC combined with cytokines, including IL-1β or tumor necrosis factor α (TNF α), decreases insulin secretion, elevates ROS, and induces apoptosis in human pancreatic β cell line 1.1B4 under high glucose conditions [72].

## 3. Dysregulated Iron Metabolism, Ferroptosis, and Diabetes

Iron, an essential micronutrient, plays a critical role in various physiological processes, including oxygen transport, energy metabolism, and enzymatic reactions [73]. However, dysregulation of iron homeostasis has been implicated in the pathogenesis of multiple diseases, including diabetes and its multi-organ complications [74,75]. Recent studies have highlighted the involvement of ferroptosis in the pathology of liver, brain, kidney, and heart diseases, as demonstrated in Figure 1 [76,77,78]. Given the central role of chronic oxidative stress and dysregulated lipid metabolism in diabetes, ferroptosis may play a vital role in the pathogenesis of diabetes and its complications [79] (Figure 2).

### 3.1. Diabetic Nephropathy (DN)

The kidneys regulate systemic iron balance by tubular reabsorption. In diabetes, the accumulation of ROS and iron overload promote the occurrence of DN. STZ-induced type 1 diabetic mice show upregulated tubular injury markers including kidney injury molecule-1 (Kim-1), neutrophil gelatinase-associated lipocalin (Ngal), plasminogen activator inhibitor-1 (PAI-1), and β2-microglobulin (B2M) [80]. Patients with DN exhibit elevated serum ferroptosis markers including elevated ROS and MDA, and decreased ACSL4, prostaglandin-endoperoxide synthase 2 (PTGS2), and GPX4 [81]. There are increased ferrous iron and MDA levels in Sprague Dawley (SD) rats with DN and human kidney proximal tubular cell line HK-2 cultured under high glucose conditions, which may indicate the involvement of ferroptosis in DN [20]. There is significant iron accumulation, reduced antioxidant capacity, and massive ROS and lipid peroxidation in the kidneys of STZ-induced DBA/2J diabetic mice and HK-2 cells under high glucose [82]. High glucose induces ferroptosis in renal mesangial SV40-MES13 cells [81]. In TGF-β1-stimulated tubular cells, intracellular GSH is reduced and lipid peroxidation is enhanced, which indicates the involvement of ferroptosis [83]. Iron dextran increases diabetes-induced kidney injury with elevated oxidative stress in high-fat and high-sucrose diet-induced diabetic rats [84]. Iron-restricted diet ameliorated mitochondrial dysfunction by restoring mitochondrial respiration and improving oxidative stress and glutathione status in the kidneys of diabetic rats [85].

### 3.2. Diabetic Osteoporosis (DOP)

Bone homeostasis relies on the dynamic balance between osteoblast-mediated bone formation and osteoclast-driven bone resorption. In diabetes, this equilibrium is disrupted. The common diabetic bone diseases include osteoporosis, increased fracture risk, and poor bone healing. DOP is a leading cause of fragility fractures in individuals with diabetes. Iron overload has been reported to lead to bone loss. Dysregulated iron metabolism proteins and signaling pathways exhibit significant alterations in the DOP progression [86].

Advanced glycation end products (AGEs) and DOP patient serum inhibit osteoblast proliferation and mineralization, and promote ferroptosis in hFOB1.19 cells [87]. High glucose decreased osteogenic function in hFOB1.19 cells and increased serum iron in diabetic rats [88]. DMT1 expression in the tibias is higher in type 2 diabetic rats than in normal rats, and its deficiency enhances bone biomechanical and microstructure [89]. Mitochondrial ferritin protects against ferroptosis in hFOB1.19 cells by sequestering excess Fe^2+^ and mitigating oxidative stress [90]. Mitochondrial ferritin deficiency triggers mitophagy, exacerbating osteoblast dysfunction in DOP.

High-glucose- and high-fat-induced osteoblastic ferroptosis is associated with elevated serum ferritin and lower levels of solute carrier family 7 member 11 (SLC7A11) and GPX4 in the bone tissues of rats with DOP, and MC3T3-E1 cells exposed to high glucose and palmitic acid exhibit suppressed mineralization and ferroptosis markers [91]. High glucose induces accumulation of lipid peroxide and decreased expression of GPX4 and SLC7A11 in osteoblasts; this further impairs the osteogenic capacity [92].

Osteocytes are the most abundant cells in mineralized bone tissues, with constant communication with osteoclasts and osteoblasts. Diabetic conditions induce osteocyte death by substantial lipid peroxidation and iron accumulation [93]. Ferrostatin 1 (Fer-1), an inhibitor of ferroptosis, rescues osteocyte viability by scavenging lipid peroxides and attenuating oxidative injury under diabetic condition.

### 3.3. Diabetic Peripheral Neuropathy (DPN)

DPN, characterized by nerve axon degeneration and impaired regeneration, arises from diabetes-induced metabolic disturbances. Dietary iron-deficiency and mild inflammation promotes neuropathy in STZ-induced diabetic rats [94]. Chronic iron depletion augments peripheral nerve pathology and pro-inflammatory activity in *ob/ob* mice [95]. In contrast to iron restriction, high iron supplementation leads to a significant increase in motor nerve conduction velocities, which partly prevents the development of DPN [96]. A cross-sectional study indicates that high dietary iron intake is associated with the presence of DPN [97]. Transcriptomic analyses identify ferroptosis-related differential genes such as *DCAF7* (DDB1- and CUL4-associated factor 7), *GABARAPL1* (GABA type A receptor-associated protein-like 1), *ACSL4* (acyl-CoA synthetase long chain family member 4), *SESN2* (sestrin 2), and *RB1* (RB transcriptional corepressor 1) in DPN patients versus healthy controls, highlighting ferroptosis pathway activation [98]. High glucose induces ferroptosis in Schwann cells with downregulated expression of GPX4 and SLC7A11, which is involved in the pathogenesis of DPN [99].

### 3.4. Diabetic Cognitive Impairment

Iron metabolism is involved in cerebral homeostasis, with its dysregulation implicated in neurodegenerative disorders and diabetes-associated cognitive dysfunction. Ferroptosis is a vital pathogenic pathway in diabetes-induced cognitive dysfunction through hippocampal oxidative damage. STZ-induced type 1 diabetic mice exhibit a prolonged latency period and reduced cumulative time spent in the target quadrant, elevated labile iron and lipid peroxidation (MDA and 4-hydroxynonenal), and downregulated Slc40a1 in the hippocampus, indicating the induction of ferroptosis [100]. The cortex tissues of diabetic mice also show downregulated expressions of FPN, ferritin light chain (FTL), and ferritin heavy chain 1 (FTH1) [101]. Increased Fe^2+^ concentration and expression of TFRC and decreased FTH are found in the hippocampal neurons in high-fat diet (HFD)/STZ-induced type 2 diabetic mice, and are associated with declined expressions of GPX4 and SLC7A11 [102].

### 3.5. Diabetic Cardiovascular Diseases

Iron metabolism dysregulation and ferroptosis have been recognized as important mechanisms in the pathogenesis and progress of cardiovascular diseases [103]. Diabetic cardiomyopathy (DCM) is a major contributor to increased morbidity and mortality in patients with diabetes and heart failure. Endothelial dysfunction is a critical and initiating contributor to the pathogenesis of diabetic cardiovascular complications. High glucose induces ferroptosis in human umbilical vein endothelial cells (HUVECs), as evidenced by the downregulated GPX4 expression, upregulated FTH1 expression, and increased lipid peroxidation [104]. AGEs induced ferroptosis in engineered cardiac tissues with increased levels of Ptgs2 and lipid peroxides, and decreased ferritin and SLC7A11 [105]. Diabetic mice develop hypertrophy and interstitial fibrosis with increased expression of ACSL4 and iron content, and lower expression of GPX4, indicating the activation of ferroptosis in the heart tissues [106]. Cardiac autophagy inhibition activates Nrf2-mediated ferroptosis in cardiomyocytes, thereby exaggerating the progression of cardiomyopathy associated with type 1 diabetes [107].

### 3.6. Diabetic Liver Dysfunction

Type 2 diabetes exacerbates the progression of metabolic dysfunction-associated steatotic liver disease (MASLD), the primary cause for the development of chronic liver diseases [108,109]. Hepatocellular death is one of the most important contributing factors to diabetes-related liver pathology and progression of liver damage. Diabetic mice exhibit accumulation of pro-oxidative iron and elevated lipid peroxidation, and downregulated expression of antioxidant defense molecules in the liver tissues [110]. Diabetes upregulates hepatic TfR1 expression, which enhances iron uptake, but iron supplementation normalizes its level, suggesting adaptive iron sensing [111]. Moreover, hepcidin is upregulated in diabetic liver to protect against the deleterious effects of iron on liver.

### 3.7. Diabetic Retinopathy (DR)

DR is one of the leading causes of blindness. High glucose induces higher levels of total iron, Fe^2+^, and ROS, and downregulated GPX4 expression in human retinal endothelial cells [112]. High glucose stimulates Glia maturation factor, promoting ACSL4 accumulation, which mediates lipid remodeling, drives ferroptosis in retinal pigment epithelial cells, and possibly disrupts the normal retinal physiological function [113]. Fer-1 inhibits ferroptosis in retinal tissues of rats with DR and high-glucose-exposed ARPE-19 cells [114]. NADPH oxidase is the main source of ROS. Inhibition of NADPH oxidase by Vas2870 ameliorates septic renal function injury by suppressing ferroptosis, accompanied by reduced MDA accumulation and upregulated ACSL4 expression in HFD-fed diabetic mice [115].

### 3.8. Diabetic Wounds

Iron plays a role in promoting blood clotting and facilitating the remodeling stage of wound healing. Impaired wound healing is a major complication of diabetes and involves sustained inflammation and oxidative stress at the wound sites. Diabetic murine wounds display delayed wound remodeling and deposition with significantly reduced iron level and dysregulation of iron metabolism [18]. High glucose induces elevated ROS and lipid peroxidation in the fibroblasts and vascular endothelial cells in diabetic wounds; this effect could be rescued by Fer-1, indicating the involvement of ferroptosis [116]. Senescent fibroblasts with lower expression of nuclear receptor coactivator 4 (NCOA4) in diabetic wounds are linked to impaired ferritinophagy and are resistant to ferroptosis [117].

## 4. Iron Metabolism and Ferroptosis-Targeted Therapeutics with Natural Products in Diabetes

Ferroptosis has been established as a pivotal driver in the pathogenesis of diabetes and its complications. Consequently, the inhibition of ferroptosis has emerged as a promising therapeutic strategy. Notably, the application of TCM has garnered increasing attention for its multi-target potential in diabetes treatment. In diabetes, dysregulated iron metabolism leads to iron accumulation, which promotes lipid peroxidation through the Fenton reaction, generating ROS that damage cell membranes and exacerbate organ damage [118]. Some TCM-derived natural products can modulate iron metabolism and scavenge lipid peroxidation, for instance, by chelating iron ions or enhancing the antioxidant defense system. TCM and TCM-derived natural products with the ability to regulate iron metabolism and mitigate ferroptosis have been explored for the treatment of various diseases, including diabetes and its complications [119].

### 4.1. Diabetic Nephropathy

Ferroptosis-related renal tubular lesions play important roles in the progression of diabetic kidney diseases. As illustrated in Figure 3 and Table 1, diverse natural products from TCM, including flavonoids, alkaloids, coumarins, terpenoids, lignans, anthraquinones, and polysaccharides, demonstrate therapeutic potential against DN by modulating ferroptosis-related pathways.

#### 4.1.1. Flavonoids

Total flavones from *Abelmoschus manihot* have been widely used in the treatment of chronic kidney disease. They mitigated renal tubular injury in diabetic rats and AGEs-exposed NRK-52E cells by reducing iron deposition and lipid peroxidation [120]. Vitexin suppresses ferroptosis in high-glucose-exposed HK-2 cells and diabetic rat renal tissues via decreasing ROS, Fe^2+^, and MDA levels while increasing GSH content [121]. Glabridin, a prenylated isoflavan from *Glycyrrhiza glabra* (licorice), ameliorates diabetes by repressing ferroptosis in diabetic rat kidneys and glucose-exposed NRK-52E cells [122,123]. Quercetin inhibits ferroptosis by downregulating TfR1, activating Nrf2, and upregulating GPX4 and SLC7A11 in diabetic mice and high-glucose-exposed HK-2 cells [124,125].

#### 4.1.2. Terpenoids

Tanshinone IIA, a diterpene quinone, inhibits ferroptosis in high-glucose-exposed mouse podocyte cells via lipid peroxidation suppression [126]. Ginkgolide B, a terpenoid, alleviates DN by mitigating ferroptosis in renal tissues [127]. Hederagenin, a pentacyclic triterpenoid, improves renal function by inhibiting ferroptosis in diabetic mice and high-glucose-exposed HK-2 cells [128]. Platycodin D, a triterpenoid saponin from *Platycodon grandiflorum*, suppresses ferroptosis in HK-2 cells under high glucose by GPX4 upregulation [129].

#### 4.1.3. Alkaloids

Leonurine, isolated from *Leonurus japonicus*, ameliorates DN by inducing ferroptosis in endothelial cells [130]. Berberine, derived from *Coptidis rhizome*, inhibits ferroptosis and alleviates DN in diabetic mice [131].

#### 4.1.4. Lignans

Schisandrin A, one of the lignans extracted from *Schisandra chinensis*, reduces high-glucose-induced ferroptosis in human renal glomerular endothelial cells and in the model of DN by modulating redox balance [132]. Umbelliferone, a phenylpropane compound, improves glucose and lipid metabolism and mitigates oxidative stress in type 2 diabetes [133,134]. It inhibits ferroptosis and activates the Nrf2/heme oxygenase-1 (HO-1) pathway in the kidneys of *db/db* mice and HK-2 cells under high glucose [135].

#### 4.1.5. Anthraquinones

Rhein is an anthraquinone and has been reported to attenuate DN by inhibiting ferroptosis in high-glucose-exposed renal cells [136]. Chicoric acid (CA) ameliorates ferroptosis in DN by promoting the ubiquitination of progestin and adipoQ receptor family member 3 (PAQR3), which alleviates the interaction between PAQR3 and the P110α pathway [137]. This mechanism reveals that CA exerts its therapeutic potential in DN by activating the Phosphoinositide 3-kinase/protein kinase B (PI3K/AKT) signaling pathway.

**Table 1 nutrients-17-02714-t001:** The changes of ferroptotic events in diabetic nephropathy.

Compound	Diabetic Models		Changes of Ferroptotic Biological Events			Ref.
	Animal Model	Cell Model	Iron Metabolism	Lipid Metabolism	GSH Metabolism	
Flavones	SD rats (200–220 g) induced by HFD, unilateral nephrectomy and STZ (35 mg/kg)	Murine PTEC cell line NRK-52E under AGE	Decreased nonheme iron content Decreased expression of TfR1	Decreased MDA and ROS	Increased expression of GPX4	[120]
Vitexin	Male Sprague Dawley rats induced by HFD (3 weeks)/STZ (35 mg/kg)	HK-2 cells under 50 mM glucose	Decreased Fe^2+^ content	Decreased MDA	Increased content of GSH Increased expression of SCL7A11 and GPX4	[121]
Glabridin	Male Sprague Dawley rats induced by HFD (3 weeks)/STZ (40 mg/kg, 10 days)	NRK-52E rat renal tubular epithelia cells under 30 mM glucose	Decreased iron content in kidney tissue Decreased expression of TfR1	Decreased MDA	Increased content of superoxide dismutase (SOD) and GSH Increased expression of SLC7A11, solute carrier family 3 member 2 (SLC3A2), catalase (CAT), and GPX4	[123]
Quercetin	/	HK-2 cells under 30 mM glucose	Decreased iron content Decreased TfR1 Increased FTH1	Decreased MDA and 4-HNE	Increased content of GSH Increased expression of SCL7A11 and GPX4	[124,125]
Tanshinone IIA	10-week-old *db/db* mice	Mouse glomerular podocyte MPC5 cells under 30 mM glucose	Decreased Fe^2+^ content	Decreased ROS and MDA	Increased GSH	[126]
Ginkgolide B	C57BL/KsJ *db/db* mice	Mouse renal podocyte MPC5 under 25 mM glucose	Decreased expression of TfR1 Increased expression of FTH1	/	Increased expression of GPX4	[127]
Hederagenin	6–8-week-old male C57BL/6J mice induced by STZ (50 mg/kg)	HK-2 cells under 25 mM glucose		Decreased lipid ROS and MDA	Increased expression of GPX4	[128]
Platycodin D	/	Human proximal renal tubule cell line HK-2 under 30 mM glucose	Decreased labile iron content Decreased expression of TfR1 Increased expression of FTH1	Decreased lipid peroxide and MDA Decreased expression of ACSL4	Increased content of GSH Increased expression of SCL7A11	[129]
Leonurine	6-week-old C57BL/6 mice induced by STZ (50 mg/kg, 5 days) and HFD (2 weeks)	Human umbilical vein endothelial cells (HUVECs) under 30 mM glucose	Increased expression of FTH1 and FTL Decreased iron content	Decreased MDA	Increased GPX4 and Nrf2 Increased GSH content	[130]
Berberine	8-week-old male C57BL/6J mice induced by STZ (65 mg/kg)	HK-2 cells under 5.5 mM glucose	Decreased expression of FTH1 Decreased iron content	Decreased MDA	Increased expression of GPX4	[131]
Schisandrin A	5–6-week-old C57BL/6 mice induced by HFD (12 weeks)/STZ (30 mg/kg, 7 days)	Human renal glomerular endothelial cells under 20 mM glucose	Decreased iron content	Decreased MDA	Increased SOD, CAT, and GSH Increased expression of GPX4	[132]
Umbelliferone	10-week-old C57BLKS/J *db/db* male mice	Human proximal renal tubule cell line HK-2 under 30 mM glucose	/	Decreased expression of ACSL4	Increased expression of GPX4	[135]
Rhein	6–8-week-old male C57BL/6J mice induced by STZ (50 mg/kg, 5 days)	Mouse glomerular podocyte MPC5 cells under 30 mM glucose	Decreased Fe^2+^ content Decreased expression of TfR	Decreased MDA	Increased expression of GPX4 and SLC7A11 Increased GSH and SOD content	[136]
Chicoric acid	C57BL/6 mice (5–6 weeks, 18–20 g) induced by HFD and STZ (30 mg/kg)	NRK-52E cells stimulated with 20 mmol/L d-glucose	Decreased iron concentration	/	Increased GSH activity and GPX4 expression	[137]
Rosa laevigata Michx. polysaccharide	C57BL/6 mice (20–22 g) induced by high-glucose and high-fat (HGHF) (8 weeks) STZ (30 mg/kg)	/	Decreased expression of transferrin and Steap3	Decreased ROS and 4-HNE	Increased expression of GPx4	[138]

#### 4.1.6. Polysaccharides

Polysaccharide from *Rosa laevigata* Michx. significantly ameliorates renal injury, inflammation, and oxidative stress in DN mice by inhibiting ferroptosis and PI3K/AKT pathway-mediated apoptosis, as well as modulating tryptophan metabolism [138]. RLP exerts therapeutic potential in DN by regulating metabolic pathways and suppressing cell death mechanisms.

The above natural products show significant potential to ameliorate DN by targeting ferroptosis. Their mechanisms include regulating iron homeostasis, reducing lipid peroxidation, and activating key signaling pathways such as Nrf2, GPX4, and PI3K/AKT. Further research is warranted to explore their therapeutic applications and underlying molecular mechanisms.

### 4.2. Diabetic Osteoporosis

Natural products targeting ferroptosis regulation have emerged as promising therapeutic agents to attenuate DOP by modulating iron-dependent cell death mechanisms and restoring bone homeostasis. The structure and information of natural products that inhibit ferroptosis to attenuate DOP are included in Figure 4 and Table 2. Poliumoside belongs to phenylpropanoid glycoside and alleviates type 2 diabetes-related osteoporosis by suppressing ferroptosis through activation of the Nrf2/GPX4 pathway, enhancing bone mineral density and mitigating oxidative stress [139]. Asperosaponin VI (AVI) alleviates osteoblast ferroptosis and DOP in mice by restoring GPX4 expression [140].

**Table 2 nutrients-17-02714-t002:** The changes of ferroptotic events in diabetic osteoporosis.

Compound	Diabetic Models		Changes of Ferroptotic Biological Events			Ref.
	Animal Model	Cell Model	Iron Metabolism	Lipid Metabolism	GSH Metabolism	
Poliumoside	C57BL/6 mice (25 ± 2 g) induced by HGHF combined with STZ (minimal dosage)	Bone mesenchymal stem cells (BMSCs) cultured in HGHF conditions	/	Decreased lipid peroxidation	Increased GSH levels	[139]
Asperosaponin VI	C57BL/6J mice induced by HFD and STZ (25 mg/kg)	Primary osteoblasts treated with high glucose and palmitic acid (HGPA)	Increased iron accumulation	Decreased lipid peroxidation	Increased expression of GPX4	[140]

### 4.3. Diabetic Cardiomyopathy

Emerging research has identified ferroptosis as a critical pathological process in DCM, prompting the exploration of natural products as potential therapeutic agents. As summarized in Figure 5 and Table 3, several bioactive products demonstrate ferroptosis-inhibiting effects in DCM. Sulforaphane is abundant in cruciferous vegetables (e.g., broccoli, cabbages, and cauliflower), and there has been growing attention in studying its effects against diabetes and diabetic complications [141]. It decreases MDA level, Ptgs2 expression, and labile iron, and increases SLC7A11, ferritin, and GSH levels in heart tissues of diabetic mice and primary cells cultured with AGEs [105]. Salidroside is the main component of the traditional Chinese herb *Rhodiola roseal*. Remarkably, pharmacological study has demonstrated that salidroside has a protective effect against diabetes-induced cardiac dysfunction [142]. Further, it protects against diabetic myocardial damage by inhibiting ferroptosis, reducing iron overload in diabetic mice [143]. Isorhapontigenin is an analog of resveratrol belonging to the family of stilbenes. Evidence shows that resveratrol fights diabetes and its complications in various types of diabetic models [144]. It protects cardiac microvasculature in diabetes by suppressing mitochondria-associated ferroptosis [145]. Gingerol is a kind of polyphenol compound extracted from ginger. Previous research indicates that 6-gingerol attenuated renal damage in diabetic rats [146]. It protects DCM by attenuating ferroptosis with decreasing cardiac iron content and upregulating GPX4 expression in the heart tissues of HFD/STZ-induced diabetic mice and high-glucose-exposed H9C2 cardiomyocytes [106]. Curcumin, a major polyphenolic compound in turmeric (*Curcuma longa*) rhizomes, is found to reduce glucose-induced myocardial cell damage and inhibit high-glucose-induced ferroptosis in cardiomyocytes [147].

### 4.4. Diabetic Pancreas Injury

In diabetes, pancreatic β cells are exposed to various stressors such as chronic hyperglycemia, oxidative stress, and pro-inflammatory cytokines, leading to progressive dysfunction. Inhibiting ferroptosis may protect pancreatic β cell function, which is crucial for glucose homeostasis, and ultimately alleviate diabetes and its complications [148]. The structure and information of natural products that inhibit ferroptosis to attenuate diabetic pancreas injury are included in Figure 6 and Table 4. High glucose triggers ferroptosis in mouse pancreatic β cells, further impairing insulin secretion [149]. Resveratrol, a polyphenol compound, inhibits acrolein-induced ferroptosis in pancreatic β cells to improve insulin secretion dysfunction [149]. Quercetin is a kind of flavonoid, and more importantly, it is found to have the effect of regulating iron metabolism [150]. It ameliorates ferroptosis by increasing antioxidant capacity with higher GSH content and GPX4 expression, while decreasing oxidative stress with lower iron deposition and lipid peroxide in pancreatic β cells; this delays the progression and development of type 2 diabetes [151]. Cryptochlorogenic acid is a bioactive compound derived from mulberry leaf. It is reported that cryptochlorogenic acid decreases iron deposits in the pancreatic tissue of diabetic rats. It also decreases TfR1 expression, activates Nrf2, and increases GPX4 level in the pancreatic tissue of diabetic rats [152]. Hispidin is a polyphenol compound isolated from *Phellinus linteus*. This polyphenol protects mouse pancreatic β cells under high glucose against ferroptosis by decreasing intracellular Fe^2+^, ROS, and MDA, and increasing GSH content [153].

### 4.5. Diabetic Recognitive Impairment

Recent advancements have revealed the pivotal role of ferroptosis in the progression of diabetic cognitive impairment, driving interest in natural products as effective interventions. The structure and information of natural products that inhibit ferroptosis to attenuate diabetic recognitive impairment are included in Figure 7 and Table 5. Sinomenine, an alkaloid from *Sinomenium acutum*, exerts anti-ferroptosis effects. In HT-22 hippocampal neuronal cells, it inhibits ferroptosis induced by both erastin and high glucose conditions. Furthermore, it protects hippocampal neurons against ferroptosis by activating Nrf2/HO-1 signaling pathways, thereby mitigating oxidative stress and iron overload in type 2 diabetic rats [154]. Dihydromyricetin, a kind of flavonoid compound, has been reported to alleviate hippocampal ferroptosis in type 2 diabetic rats with cognitive impairment [155]. Its mechanism involves restoring redox homeostasis and suppressing lipid peroxidation. Dendrobine, a pyrrolizidine-derivative alkaloid from *Dendrobium officinale*, demonstrated a neuroprotective effect in diabetic encephalopathy. It attenuates ferroptosis by activating Nrf2/GPX4, which enhanced cellular antioxidant capacity and reduced iron-dependent oxidative damage in the hippocampus [156].

### 4.6. Diabetic Retinopathy

Growing evidence underscores the involvement of ferroptosis in the pathogenesis of DR, sparking investigations into natural products as potential therapeutic strategies. The structure and information of natural products that induce ferroptosis in DR are included in Figure 8 and Table 6. Astragaloside IV is a bioactive saponin extracted from *Astragalus membranaceus*. It protects against retinal iron overload toxicity through reducing iron deposition [157]. The pharmacological actions of astragaloside IV are reported mainly from the inhibitory effect on ferroptosis and the antioxidant and free radical-scavenging activities [158,159]. Importantly, it has been reported to inhibit ferroptosis by increasing the expression of GPX4 and enhancing Nrf2 activity in high-glucose-exposed ARPE-19 retinal pigment epithelia cells [160]. 1,8-Cineole, a major volatile component of aromatic plants, has shown therapeutic promise in DR. It ameliorates DR by inhibiting ferroptosis in both high-glucose-exposed retinal pigment epithelium cells and retinal tissues of diabetic mice [161].

### 4.7. Diabetic Peripheral Neuropathy

The identification of ferroptosis as a key contributor to DPN has led to an increased focus on natural products as viable treatment options. The structure and information of natural products that induce ferroptosis in DPN are included in Figure 9 and Table 7. High glucose induces Schwann cell injury, contributing to DPN. Honokiol is isolated from *Magnolia officinalis*, and was found to protect Schwann cells from high-glucose-induced damage by inhibiting ferroptosis and activating the AMPK/silent information regulator 1 (SIRT1)/peroxisome proliferator-activated receptor gamma coactivator-1α (PGC-1α) pathway [162]. High glucose increases intracellular iron accumulation and lipid peroxidation, leading to ferroptosis, while honokiol restores the expression of ferroptosis suppressors SLC7A11 and GPX4. Additionally, in an STZ-induced diabetic rat model, honokiol treatment significantly improves sciatic nerve conduction velocity and thermal sensitivity, demonstrating its therapeutic potential for DPN [162].

### 4.8. Diabetic Atherosclerosis

Ferroptosis has been increasingly recognized as a critical factor in the development of diabetic atherosclerosis, prompting the exploration of natural products as potential therapeutic agents. The structure and information of natural products that induce ferroptosis in diabetic atherosclerosis are included in Figure 9 and Table 7. Hydroxysafflor yellow A, a single chalcone glycoside and the active ingredient of *Carthamus tinctorius* L, has been shown to attenuate high-glucose-induced human umbilical vein endothelial cell dysfunction and inhibit ferroptosis in ischemia/reperfusion injury [163,164]. It reduces atherosclerotic plaque formation in type 2 diabetic mice with atherosclerosis and inhibits human umbilical vein endothelial cell ferroptosis through upregulating GPX4 expression and decreasing MDA and iron content [165].

**Table 7 nutrients-17-02714-t007:** The changes of ferroptotic events in diabetic peripheral neuropathy and diabetic atherosclerosis.

Compound	Diabetic Models		Changes of Ferroptotic Biological Events			Ref.
	Animal Model	Cell Model	Iron Metabolism	Lipid Metabolism	GSH Metabolism	
Honokiol	6–8-week-old male SD rats induced by STZ (60 mg/kg)	Rat Schwann cell line RSC96 under 150 mM glucose	Decreased Fe^2+^ Decreased expression of TfR1 Increased expression of FTH	Decreased MDA and ROS	Increased expression of GPX4, NRF2, and SLC7A11 Increased GSH	[162]
Hydroxysafflor yellow A	*ApoE*^−/−^-deficient C57BL/6 mice induced by HFD (4 weeks)/STZ (30 mg/kg, 3 days)	Human umbilical vein endothelial cells under 33.3 mM glucose	Decreased iron	Decreased MDA and ROS Decreased expression of ACSL4	Increased expression of GPX4, GSH, and SLC7A11	[165]

## 5. Conclusions

In this comprehensive review, we have explored the intricate interplay between iron metabolism, ferroptosis, and diabetes, elucidating the significant regulatory role of natural products within this context.

The relationship between iron and diabetes is highly complex and multifarious. Dysregulation of iron homeostasis has been identified as a pivotal factor contributing to the onset and progression of diabetes. Excessive iron accumulation can trigger oxidative stress, inflammation, and impairment of insulin signaling, ultimately exacerbating the diabetic condition. A comprehensive understanding of these underlying mechanisms provides a solid foundation for the exploration of novel therapeutic approaches. Furthermore, the association between iron and diabetic complications underscores the critical importance of maintaining iron homeostasis. Iron-mediated oxidative damage has been implicated in the pathogenesis of diabetic complications. Targeting iron metabolism holds the potential to attenuate the development and progression of these complications, thereby offering new prospects for enhancing the quality of life of diabetic patients.

The investigation of iron metabolism and ferroptosis-targeted therapeutic applications in diabetes and its complications has unveiled the promising potential of natural products, particularly those derived from TCM. These natural products exhibit diverse mechanisms of action, encompassing the regulation of iron absorption, transport, and storage, as well as the modulation of ferroptosis pathways. However, despite these promising findings, several challenges persist. The precise mechanisms of action of many natural products require further clarification, and their safety and efficacy in clinical settings necessitate rigorous evaluation. Additionally, the development of standardized extraction and formulation methods for these natural products is of utmost importance to ensure their consistent quality and therapeutic efficacy.

In conclusion, the regulatory role of natural products in iron metabolism and ferroptosis holds great promise for diabetes treatment. Future research efforts should be directed towards addressing the existing challenges, conducting more in-depth preclinical and clinical investigations, and exploring the synergistic effects of different natural products. With sustained efforts, these natural products may provide novel and effective therapeutic options for the management of diabetes and its complications, contributing to improved health outcomes for patients.

## Figures and Tables

**Figure 1 nutrients-17-02714-f001:**
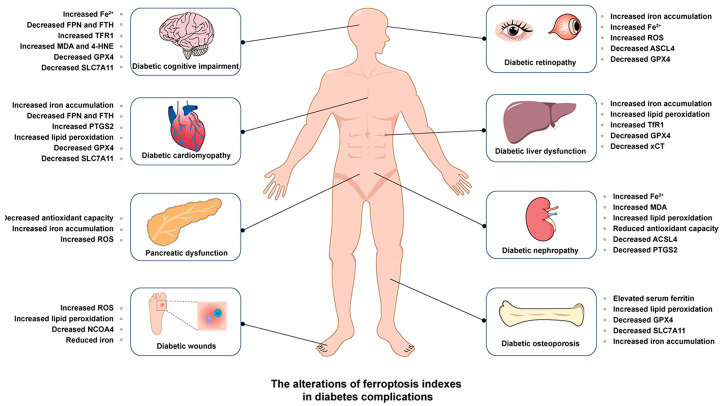
Ferroptosis markers in diabetic complications. This figure illustrates the changes in ferroptosis-related markers across various diabetic complications using a schematic representation of the human body.

**Figure 2 nutrients-17-02714-f002:**
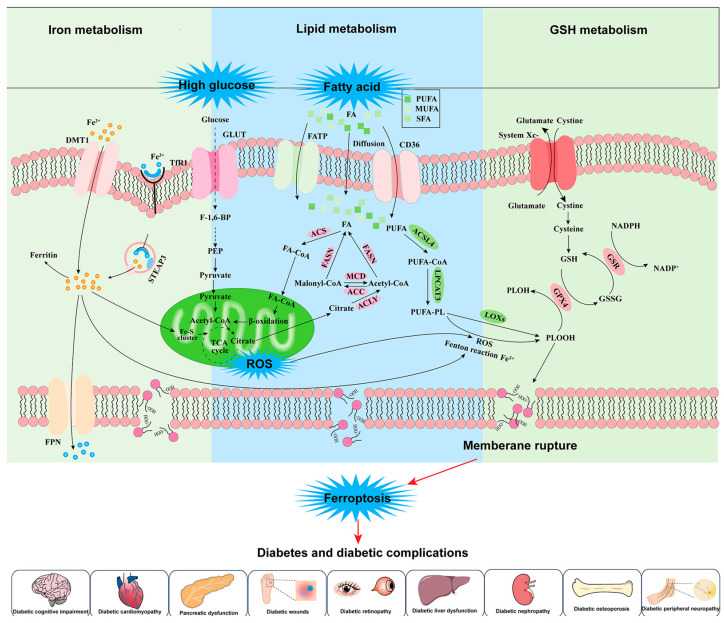
The mechanism of ferroptosis and the activation of ferroptosis in diabetes and diabetic complications. Iron metabolism: Ferrous iron (Fe^2+^) enters the cytosol via DMT1-mediated active transport across the plasma membrane. Concurrently, transferrin-bound ferric iron (Fe^3+^) is internalized through transferrin receptor 1 (TfR1)-mediated endocytosis. Within endocytic vesicles, Fe^3+^ is reduced to Fe^2+^ by six-transmembrane epithelial antigen of prostate 3 (STEAP3) and subsequently exported to the cytosol via FPN1. Cytosolic Fe^2+^ undergoes three distinct metabolic fates: (1) sequestration by ferritin for inert storage; (2) mitochondrial import for iron–sulfur cluster (Fe-S cluster) biogenesis, where these cofactors serve as electron carriers supporting the tricarboxylic acid (TCA) cycle and oxidative phosphorylation; and (3) participation in the Fenton reaction, catalyzing ROS generation through oxidation of Fe^2+^ to Fe^3+^ with concomitant production of highly reactive hydroxyl radicals (·OH). This iron-catalyzed process constitutes a core mechanism of ferroptosis, an iron-dependent form of regulated cell death. The precise coordination of this metabolic network maintains cellular iron homeostasis, with dysregulation leading to oxidative stress or iron metabolism disorders. Glucose metabolic flux: Glucose enters the cytoplasm through facilitative diffusion mediated by glucose transporters (GLUTs). Intracellular glucose then undergoes glycolysis, where it is initially phosphorylated to glucose-6-phosphate (G6P) and subsequently converted to fructose-1,6-bisphosphate (F-1,6-BP) under the regulation of phosphofructokinase-1 (PFK-1). The glycolytic pathway ultimately yields phosphoenolpyruvate (PEP), which is transformed into pyruvate via pyruvate kinase (PK) catalysis. Pyruvate is then transported across the mitochondrial membrane into the matrix through the mitochondrial pyruvate carrier, where the pyruvate dehydrogenase complex (PDH) oxidatively decarboxylates it to form acetyl-CoA. This key metabolite subsequently condenses with oxaloacetate to generate citrate, thereby initiating the TCA cycle. Lipid metabolism: saturated fatty acids (SFAs) primarily enter cells through passive diffusion, whereas monounsaturated fatty acids (MUFAs) and polyunsaturated fatty acids (PUFAs) rely on active uptake mediated by membrane transport proteins CD36 and FATP; following cellular entry, fatty acids are activated to fatty acyl-CoA (FA-CoA) by acyl-CoA synthetase (ACS) and subsequently undergo β-oxidation in mitochondria to generate acetyl-CoA (Acetyl-CoA), which serves as a substrate for the tricarboxylic acid (TCA) cycle; the citrate produced by the TCA cycle can be reconverted to acetyl-CoA by ATP-citrate lyase (ACLY), and this acetyl-CoA can either be used for de novo fatty acid synthesis catalyzed by fatty acid synthase (FASN) or converted to malonyl-CoA (malonyl-CoA) by acetyl-CoA carboxylase (ACC), with malonyl-CoA being reversibly converted back to acetyl-CoA via malonyl-CoA decarboxylase (MCD); notably, PUFAs are activated to PUFA-CoA by ACSL4 and subsequently incorporated into membrane phospholipids as PUFA-PL by lysophosphatidylcholine acyltransferase 3 (LPCAT3), where they are oxidized to phospholipid hydroperoxides (PLOOHs) by ROS and lipoxygenases (LOXs); while GPX4 can reduce PLOOHs to phospholipid alcohols (PLOHs), excessive accumulation of PLOOHs leads to membrane integrity disruption and triggers ferroptosis. GSH metabolism: glutamate and cystine enter cells through the system Xc- antiporter in a 1:1 exchange ratio, wherein cystine is reduced to cysteine, which subsequently serves as a key substrate for GSH synthesis; as the primary antioxidant, GSH, catalyzed by GPX4, reduces lipid hydroperoxides (LOOHs) to their corresponding alcohols (LOHs) while being oxidized to oxidized glutathione (GSSG); GSSG is then regenerated back to GSH through the catalytic action of glutathione reductase (GSR) utilizing the reducing power of NADPH, with concomitant oxidation of NADPH to NADP^+^, thereby completing the glutathione cycle.

**Figure 3 nutrients-17-02714-f003:**
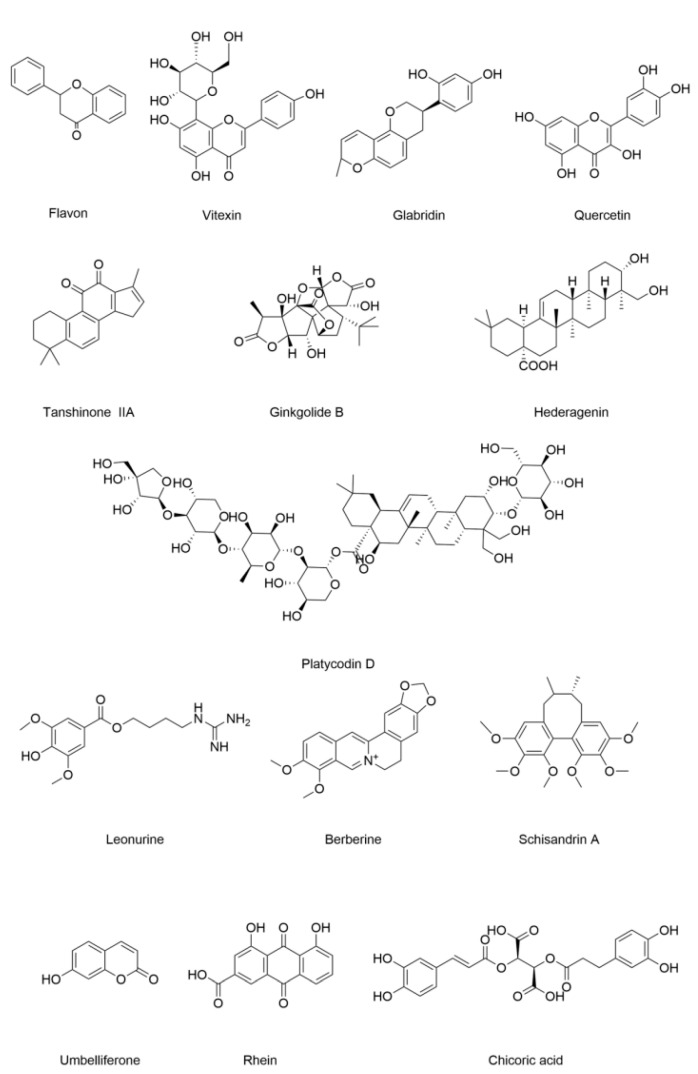
Structural formulas of compounds associated with ferroptotic events in diabetic nephropathy.

**Figure 4 nutrients-17-02714-f004:**
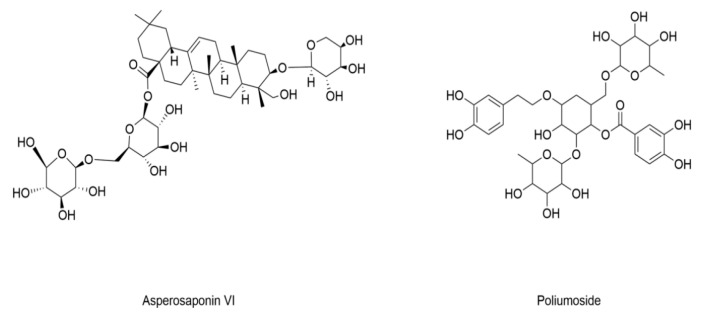
Structural formulas of compounds associated with ferroptotic events in diabetic osteoporosis.

**Figure 5 nutrients-17-02714-f005:**
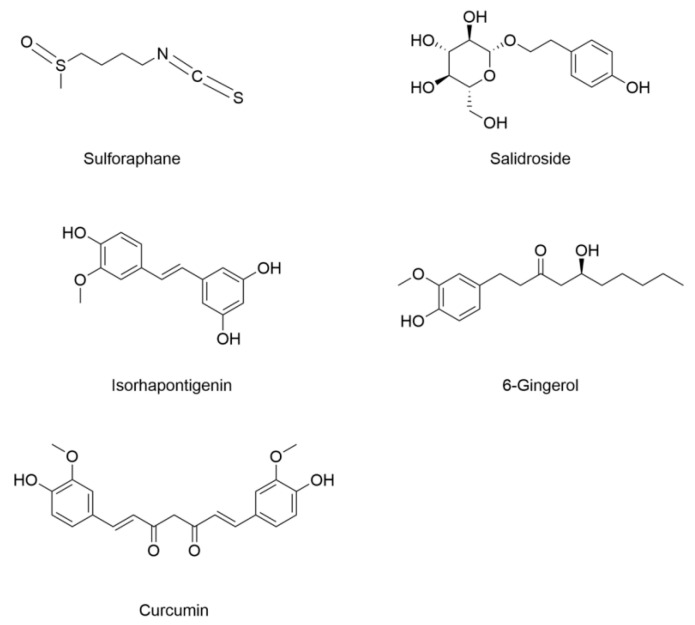
Structural formulas of compounds associated with ferroptotic events in diabetic cardiomyopathy.

**Figure 6 nutrients-17-02714-f006:**
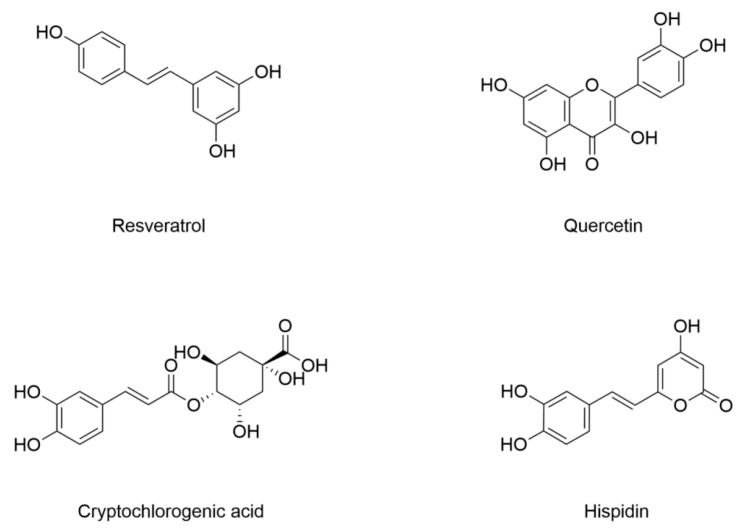
Structural formulas of compounds associated with ferroptotic events in diabetic pancreas impairment.

**Figure 7 nutrients-17-02714-f007:**
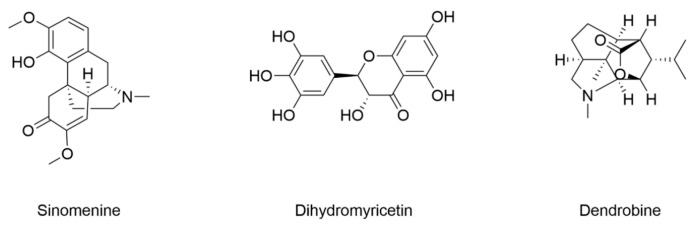
Structural formulas of compounds associated with ferroptotic events in diabetic cognitive impairment.

**Figure 8 nutrients-17-02714-f008:**
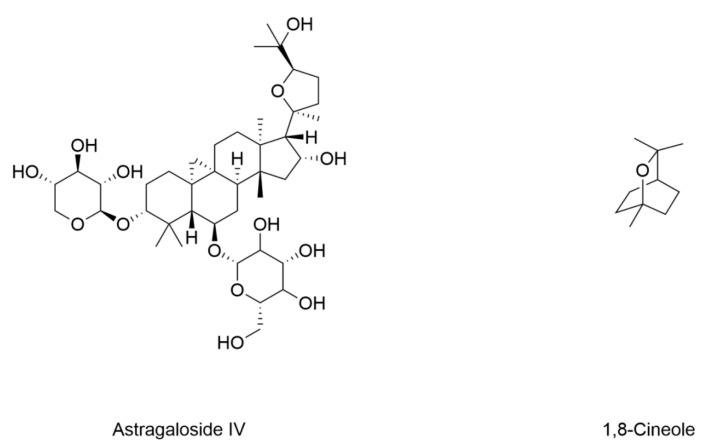
Structural formulas of compounds associated with ferroptotic events in diabetic retinopathy.

**Figure 9 nutrients-17-02714-f009:**
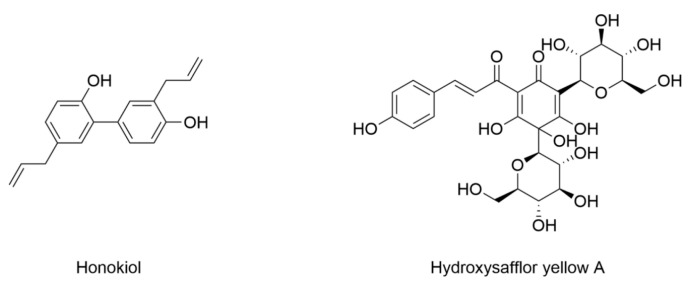
Structural formulas of compounds associated with ferroptotic events in diabetic peripheral neuropathy and diabetic atherosclerosis.

**Table 3 nutrients-17-02714-t003:** The changes of ferroptotic events in diabetic cardiomyopathy.

Compound	Diabetic Models		Changes of Ferroptotic Biological Events			Ref.
	Animal Model	Cell Model	Iron Metabolism	Lipid Metabolism	GSH Metabolism	
Sulforaphane	8-week-old male C57BL/6 mice induced by HFD (3 months)/STZ (100 mg/kg)	Primary cardiac cells from FVB pups	Increased expression of ferritin Decreased labile iron content	Decreased MDA	Decreased expression of prostaglandin-endoperoxide synthase (Ptsg2) Increased GSH content and GSH/GSSG ratio Increased expression of SLC7A11	[105]
Salidroside	8-week-old male BKS-Leprem2cd479/GPT (*db/db*) mice	/	Decreased serum transferrin and iron content	/	Increased expression of GPX4	[143]
Isorhapontigenin	C57BLKS/J *db/db* mice induced by long-term diabetes (24 weeks)	Primary cardiac microvascular endothelial cells (CMECs) treated with high glucose (HG) and free fatty acids (FFAs)	Decreased ferrous iron content	Decreased lipid peroxidation	Increased GSH content and GPX4 expression	[145]
6-Gingerol	4–5-week-old male C57BL/6 mice induced by HFD (4 weeks)/STZ (50 mg/kg, 5 days)	H9c2 rat cardiomyoblast cells under 33 Mm glucose	Decreased iron content	Decreased MDA	Increased expression of GPX4 and SOD	[106]
Curcumin	2-month-old male New Zealand rabbits induced by STZ (80 mg/kg)	Rat H9C2 cardiomyocytes under 30 mM glucose	/	/	Increased expression of GPX4 and HO-1	[147]

**Table 4 nutrients-17-02714-t004:** The changes of ferroptotic events in diabetic pancreas impairment.

Compound	Diabetic Models		Changes of Ferroptotic Biological Events			Ref.
	Animal Model	Cell Model	Iron Metabolism	Lipid Metabolism	GSH Metabolism	
Resveratrol	/	Min6 cells under 25 mM acrolein	/	Decreased MDA	Increased expression of GPX4 Decreased ACSL4	[149]
Quercetin	7–8-week-old C57BL/6 mice induced by HFD (4 months)/STZ (50 mg/kg, 5 days)	INS-1 cells under 11.1 Mm glucose	Decreased iron deposition and expression of FTL	Decreased MDA	Increased GSH content Increased expression of SOD and GPX4 Decreased expression of xCT	[151]
Cryptochlorogenic acid	SD rats (250–270 g) induced by STZ (50 mg/kg)	INS-1 cells under 50 mM glucose	Decreased expression of TfR1 Decreased iron	Activated Nrf2 Decreased MDA	Increased expression of GPX4 Increased content of GSH Decreased content of GSSG	[152]
Hispidin	/	Min6 cells under 10 mM glucose	Decreased Fe^2+^	Decreased ROS and MDA	Increased content of GSH	[153]

**Table 5 nutrients-17-02714-t005:** The changes of ferroptotic events in diabetic cognitive impairment.

Compound	Diabetic Models		Changes of Ferroptotic Biological Events			Ref.
	Animal Model	Cell Model	Iron Metabolism	Lipid Metabolism	GSH Metabolism	
Sinomenine	8-week-old male SD rats induced by HGHF (4 weeks)/STZ (25 mg/kg)	Mouse hippocampal neuron cell line HT22 under 30 mM glucose	Decreased Fe^2+^	Decreased MDA and ROS	/	[154]
Dihydromyricetin	6–8-week-old male SD rats induced by HFD (4 weeks)/STZ (30 mg/kg)	/	Decreased Fe^2+^	Decreased MDA and ROS Decreased expression of ACSL4	Increased expression of GPX4 Increased GSH	[155]
Dendrobine	12-week-old male *db/db* mice	Mouse hippocampal neuron cell line HT22 under 400 μg/mL AGEs	Decreased iron content Decreased expression of TfR1 Increased expression of FPN and FTH	Decreased MDA	Increased expression of GPX4, Nrf2, HO-1, and NQO1 Increased SOD and GSH	[156]

**Table 6 nutrients-17-02714-t006:** The changes of ferroptotic events in diabetic retinopathy.

Compound	Diabetic Models		Changes of Ferroptotic Biological Events			Ref.
	Animal Model	Cell Model	Iron Metabolism	Lipid Metabolism	GSH Metabolism	
Astragaloside IV	/	Human retinal pigment epithelium ARPE-19 cells under 25 mM glucose	/	Decreased lipid peroxidation and ROS Activated Nrf2	Increased expression of GPX4 and GCLC	[160]
1,8-Cineole	8-week-old male C57BL/6 mice induced by HFD (8 weeks) STZ (55 mg/kg)	ARPE-19 cells under 25 mM glucose	Decreased Fe^2+^	Decreased MDA	Increased GSH Increased expression of GPX4 and FSP-1	[161]

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
