# Peer review of "Natural Products as Modulators of Iron Metabolism and Ferroptosis in Diabetes and Its Complications"

_nutrients, 2025, doi:10.3390/nu17162714_

Round 1
Reviewer 1 Report
Comments and Suggestions for Authors
The review tackles a timely topic, but several changes should be reviewed, as they are:
-
Mechanistic errors in Figure 1 caption (iron handling / ROS). The caption says: “Both forms are subsequently converted into ferritin. Ferritin contributes to the generation of ROS through the Fenton reaction and the TCA cycle.” Biochemically, ferritin sequesters iron and limits the labile Fe²⁺ pool; the Fenton reaction involves labile Fe²⁺, not ferritin-bound iron. Also, ferritin is not a ROS-generating enzyme and is not a TCA-cycle participant. Please correct the iron-trafficking sequence (Tf-TFR1 endocytosis → endosomal STEAP3 reduction → DMT1 export to cytosol → labile iron pool → utilization/storage/export) and rephrase the ROS section accordingly. The phrase “both forms are subsequently converted into ferritin” is also inaccurate; only a fraction of cytosolic Fe enters ferritin.
-
Effect direction on ferroptosis is repeatedly inverted. In several places the text says natural products “inducing ferroptosis to attenuate” diabetic complications (e.g., DOP and pancreas sections), when the described studies actually inhibit ferroptosis to protect tissues. Examples: “natural products that inducing ferroptosis to attenuate DOP” ; similar wording appears for pancreatic injury and cognitive impairment sections . Please correct these to “inhibiting ferroptosis to attenuate…”.
-
Insulin sensitivity vs ferritin—internal inconsistency. The manuscript first frames higher ferritin as a risk factor for T2D, aligned with insulin resistance, yet later states “insulin sensitivity is positively related to serum ferritin and transferrin saturation” (and negatively related to transferrin) . This conflicts with the broader narrative of iron overload associating with insulin resistance. Either justify with the cited study’s context (population, adjustment, non-linear/U-shaped effects) or reword to avoid misinterpretation.
Author Response
Reviewer#1
Comment 1: Mechanistic errors in Figure 1 caption (iron handling / ROS). The caption says: “Both forms are subsequently converted into ferritin. Ferritin contributes to the generation of ROS through the Fenton reaction and the TCA cycle.” Biochemically, ferritin sequesters iron and limits the labile Fe²⁺ pool; the Fenton reaction involves labile Fe²⁺, not ferritin-bound iron. Also, ferritin is not a ROS-generating enzyme and is not a TCA-cycle participant. Please correct the iron-trafficking sequence (Tf-TFR1 endocytosis → endosomal STEAP3 reduction → DMT1 export to cytosol → labile iron pool → utilization/storage/export) and rephrase the ROS section accordingly. The phrase “both forms are subsequently converted into ferritin” is also inaccurate; only a fraction of cytosolic Fe enters ferritin.
Response to comment 1: We regret the error in the previous version. The issue has been thoroughly addressed in lines 286-297 of the revised document.
Comment 2: Effect direction on ferroptosis is repeatedly inverted. In several places the text says natural products “inducing ferroptosis to attenuate” diabetic complications (e.g., DOP and pancreas sections), when the described studies actually inhibit ferroptosis to protect tissues. Examples: “natural products that inducing ferroptosis to attenuate DOP” ; similar wording appears for pancreatic injury and cognitive impairment sections . Please correct these to “inhibiting ferroptosis to attenuate…”.
Response to comment 2: We sincerely apologize for this oversight caused by our negligence. The corrections have been comprehensively implemented. They are located at lines 519-520, line 557, and line 585.
Comment 3: Insulin sensitivity vs ferritin—internal inconsistency. The manuscript first frames higher ferritin as a risk factor for T2D, aligned with insulin resistance, yet later states “insulin sensitivity is positively related to serum ferritin and transferrin saturation” (and negatively related to transferrin). This conflicts with the broader narrative of iron overload associating with insulin resistance. Either justify with the cited study’s context (population, adjustment, non-linear/U-shaped effects) or reword to avoid misinterpretation.
Response to comment 3: We have reformulated this issue in lines 148-151 to ensure consistency. Through careful examination of both articles, we have identified distinct emphases and variations in their research subjects.
Reviewer 2 Report
Comments and Suggestions for Authors
The authors have tackled an interesting topic in their work. The combination of natural products, ferroptosis, diabetes, and its complications is exciting.
The introduction adequately introduces the topic. However, although this is not a systematic review, the authors should include a few sentences about their methodology. I'm referring to how they collected the literature for this article, which databases they reviewed, what keywords they used, and what timeframe they assumed for the published articles. Did they exclude any items, for example, only those available in English, or exclude reviews? It would be helpful here.
In lines 94-95, the authors should emphasize that these are acute-phase proteins when mentioning ceruloplasmin, transferrin, and albumin. In states of chronic inflammation, these proteins do not always reflect iron metabolism. DM is undoubtedly a state of chronic inflammation combined with increased oxidative stress.
Figure 1 is the only figure presenting a diagram of the connections between oxidative stress, iron, lipids, and ferroptosis. Although an interesting and extensive description accompanies it, the figure itself is unintuitive. I suggest that the researchers add numbers to the figure and refer to them in the description. It will allow the reader to navigate the figure more efficiently.
Since the article generally contains many abbreviations, I suggest that the authors add a list of abbreviations at the end.
I have no objections to the citations; the authors correctly reference the publications and provide appropriate commentary.
The conclusions are sound, and the authors appropriately mention certain limitations, stating that the mechanisms of action of many natural compounds are still unknown.
Author Response
Reviewer#2
Comment 1: The authors have tackled an interesting topic in their work. The combination of natural products, ferroptosis, diabetes, and its complications is exciting. The introduction adequately introduces the topic. However, although this is not a systematic review, the authors should include a few sentences about their methodology. I'm referring to how they collected the literature for this article, which databases they reviewed, what keywords they used, and what timeframe they assumed for the published articles. Did they exclude any items, for example, only those available in English, or exclude reviews? It would be helpful here.
Response to comment 1: It’s very important. We have supplemented the relevant content in lines 75-85 as suggested.
Comment 2: In lines 94-95, the authors should emphasize that these are acute-phase proteins when mentioning ceruloplasmin, transferrin, and albumin. In states of chronic inflammation, these proteins do not always reflect iron metabolism. DM is undoubtedly a state of chronic inflammation combined with increased oxidative stress.
Response to comment 2: We have implemented the necessary modifications regarding this issue in lines 104-107.
Comment 3: Figure 1 is the only figure presenting a diagram of the connections between oxidative stress, iron, lipids, and ferroptosis. Although an interesting and extensive description accompanies it, the figure itself is unintuitive. I suggest that the researchers add numbers to the figure and refer to them in the description. It will allow the reader to navigate the figure more efficiently.
Response to comment 3: We sincerely apologize for the previous confusing figure captions which may have caused reading difficulties. To address this issue, we have now divided the figure into four distinct sections: Iron Metabolism, Glucose Metabolic Flux, Lipid Metabolism, and GSH Metabolism. We believe this revised presentation will significantly improve readability and comprehension.
Comment 4: Since the article generally contains many abbreviations, I suggest that the authors add a list of abbreviations at the end. I have no objections to the citations; the authors correctly reference the publications and provide appropriate commentary. The conclusions are sound, and the authors appropriately mention certain limitations, stating that the mechanisms of action of many natural compounds are still unknown.
Response to comment 4: A list of abbreviations has been included based on your recommendation. We appreciate your input.